# The Aroma Composition of Koryciński Cheese Ripened in Different Temperatures

**DOI:** 10.3390/molecules27248745

**Published:** 2022-12-09

**Authors:** Jarosław Kliks, Zuzanna Białobrzycka, Martyna Krzyszkowska, Justyna Korycka-Korwek, Mateusz Ciepliński, Mariusz Kasprzak

**Affiliations:** Faculty of Biological Sciences, University of Zielona Gora, Prof. Z. Szafrana 1 Street, 65-417 Zielona Gora, Poland

**Keywords:** cheese, Korycin, GC/MS, volatiles, ripening

## Abstract

As a regional product, Koryciński cheese is one of the most important cheeses in the Podlasie region of Poland. In this study, the influence of technological processes, such as ripening, on shaping the organoleptic characteristics of cheese was determined. Korycin-type cheeses are produced from cow’s milk according to traditional technology. The ripening process lasted 466 h at 5 °C, 10 °C, and 15 °C. The aging temperature had a decisive influence on the number of esters and organic acids formed, which were analyzed by gas chromatography–mass spectrometry (GC/MS). The organoleptic properties of the cheeses were also related to the ripening temperature.

## 1. Introduction

Koryciński cheese (ser Koryciński) is an original Polish traditional cheese produced in the Podlasie district in the Korycin municipality. Since 2005, its name has been protected by the State List of Traditional Product. In 2012, it was registered by the European Commission, receiving a certificate of Protected Geographical Indication (PGI). Its production is limited exclusively to the areas of the three Podlasie province municipalities located in the Sokółka district: Korycin, Suchowola, and Janów [1]. The community of Koryciński cheese producers continues to grow, and, in 2011, united over 30 members [2]. For the production of Koryciński cheese, unpasteurized cow’s milk, rennet, and a denaturation temperature of 33–36 °C are used. The ripening of Koryciński cheese depends on the individual production conditions and lasts 2–6 weeks at room temperature, which leads to particular variations in the sensory characteristics of the final product. This unique product is the reason for the popularity of the Korycin region [3,4].

Koryciński cheese is a semi-hard cheese with a distinctive creamy texture. Many biological and chemical processes occur during cheese ripening, especially lipolysis, proteolysis, and acidification [5]. Lipolysis and proteolysis processes are usually autocatalytic, while microbial activity is responsible for acidification [6,7]. The microbial composition of raw milk consists of lactic acid bacteria (LAB), *Brevibacterium linens*, yeasts and molds, *Staphylococcus aureus* and other *Micrococcaceae*, *Pseudomonas*, *coliforms*, *Escherichia coli*, *enterococci*, *Clostridium perfringens*, and spores of anaerobic lactate-fermenting bacteria. *Pseudomonas*, *Lactococci*, and *Micrococcaceae* were the most numerous groups [8,9,10].

The volatile composition of semi-hard cheeses is one of the most substantial sensory attributes aside from taste. The most common compounds of semi-hard cheeses are esters—Ethyl hexanoate, Ethyl propanoate, and Ethyl octanoate; ketones—Acetone, Octadien –2–One, and aldehydes—Hexanal, Heptanal, Octanal, and Nonanal [11,12,13].

Currently, the profile of volatile compounds in regional cheeses is studied using headspace-solid phase microextraction (HS–SPME) techniques [14,15,16]. However, in recent years, modifications in the form of vacuum-assisted HS–SPME (Vac–HS–SPME) have also been used in the study of semi-hard cheeses, giving excellent results in the determination of short-chain volatile compounds, as noted by Sýkor et al. [17]. Solid-phase microextraction (SPME) is a classical research technique and still forms the basis for the analysis of volatile compounds [18,19]. The current trend is to determine the influence of microorganisms on the development of the sensory and aromatic properties of cheese [20,21]. The most commonly used technological modifications are fungi of the genus *Aspergillus* [22,23], *Enterococcus faecium* [24], or *Monascus* [6,25,26].

The aim of this article is to compare the volatile composition with sensory data during the ripening of Koryciński cheeses under different temperature conditions.

## 2. Results and Discussion

Figure 1 shows the moisture ratio (MR) dependence on the maturation time of the analyzed cheese. The moisture ratio decreases with the maturation time in all the samples studied. This process is dynamic for up to 60 h. The cheese’s humidity depended on the ripening temperature—the highest MR (0.88) was recorded for 5 °C and the lowest MR (0.84) for 15 °C. Koryciński cheese, which belongs to the semi-hard cheese group, releases a large amount of water during maturation. Tulyaganovich et al. [27] indicate that this is a typical feature of this type of cheese. Other researchers also observed a decrease in moisture content during the cheese’s maturation process [28,29]. The cheese’s ripening temperature is an essential parameter affecting the physicochemical properties of cheese, such as moisture, due to the properties of the casein complex and its ability to hold water [30,31,32]. Researchers analyzing the cheese’s maturation processes point out the crucial role of the temperature and humidity in shaping the quality of the final product [33,34]. As indicated by Łepecka et al. [35], in the production of semi-hard cheese, apart from external conditions such as the ripening temperature, specific species of bacteria from unpasteurized milk may be of critical importance in physicochemical properties, including moisture retention.

Figure 2 shows the sensory evaluation of Korycin cheeses matured under different conditions. In each test subject during criterion assessment, the cheese maturing at 10 °C has an advantage over the rest. A ripening temperature lowered by 5 °C has a less favorable effect on the aroma or external factors such as the appearance or consistency of the cheese. Cheeses ripened at the highest temperature (15 °C) scored the lowest on the organoleptic evaluation. Other researchers who observed the cheese’s ripening at different temperatures found that it plays a substantial role in profiling the evaluation of the final product [32,36,37]. Walsh et al. [32] indicate that cheddar maturation at 10 °C has a positive effect on its sensory characteristics, such as smell (mainly buttery, fatty, and pleasant). Many researchers point out the ripening time of the cheese has a significant influence on the course of lipolysis [38,39,40].

The aromatic profile (Table 1) of the studied cheeses consisted of 38 chemical compounds, the amount of which strongly depended on the conditions of the maturation process. There were 11 compounds from the group of hydrocarbons; 9 esters; 5 aldehydes; 5 organic acids; 2 alcohols; 2 lactones; 1 ketone; and 1 monoterpene.

No trend was observed in the hydrocarbon group to determine the effect of the temperature on the overall behavior of the entire group. Nevertheless, individual compounds underwent significant changes depending on the temperature of the cheese ripening. The greatest contribution in this group was n-dodecane, whose content in the aromatic profile decreased with the increasing temperature (16.35 µg/kg for 5 °C and 4.37 µg/kg for 15 °C). Similarly, in the studies of Bertuzzi et al. [41] and Atik et al. [42], the studied cheeses showed no significant effect compared to the other compounds.

In the ester group, a significant effect of the temperature on the behavior of all the compounds representing this group was observed. With the increasing cheese ripening temperature, a significant reduction was observed in all the esters. The most important compounds were the hexanoic acid ethyl ester and the acetanoic acid ethyl ester. A similar decrease in the ester content was reported when Regginito cheese was analyzed [43]. A study by Di Donato et al. [14] similarly confirmed a decrease in the ester content, including the key hexanoic acid ethyl ester.

Another group, whose share in the aromatic profile of the studied cheeses is relatively small, consists of aldehydes. The sum of the aldehydes ranges from 4–10 µg/kg, and their share is not statistically significantly dependent on the temperature of the cheese’s maturation process. A study by Ceruti et al. [43] reported similar aldehyde levels. Furthermore, the time or method of ripening did not significantly affect the aldehyde content in the aromatic profile of the cheeses. A study by Van Leuven et al. [44] also confirmed that the aldehyde group was quantitatively an insignificant part of the aromatic profile of the ripened cheeses.

The organic acids are the most important group giving aroma to the cheeses studied. A significant influence of the temperature on the formation of the organic acids’ content was observed—with an increase in the temperature of the ripened cheeses, an increase in the content of the organic acids in the volatile fraction was observed. The main aromatic compounds identified were hexanoic acid (26.23–108.24 µg/kg) and acetanoic acid (43.91–103.54 µg/kg). Many researchers confirm the fact that the amount of octanoid acid also increases with the number of days until the 25th week of cheese maturation [45,46,47].

The long-chain alcohols n–pentadecanol, and 1 decanol–2hexyl, had no significant effect on shaping the volatile compound profiles of the studied cheeses. Due to the chain length, the low volatility of these compounds is to be expected. Additionally, a study by Amárita et al. [48] confirmed the absence of long-chain alcohols in the cheese aromatics detected by the SPME-GC/MS technique [48].

In the group of lactones, the content of the volatile profile of 4–hexanolide was significantly dependent on the ripening temperature. An increase in the temperature resulted in a decrease in the 4–hexanolide content in the aromatic profile of cheeses. Lactones with few carbons in the ring are frequently detected compounds in the aromatic profiles of various ripened cheeses [44]. Researchers confirm that the influence of ripening on labeled lactone compounds can be significant [49,50].

## 3. Materials and Methods

### 3.1. Cheesemaking

We made Koryciński cheeses from cow’s milk from a local dairy farm. We used whole milk containing 4.19% fat, 3.5% protein, and 4.7% carbohydrates. Figure 3 shows the technological process and production conditions. Beaugel 50 and Beaugel 150 (0.58 g CaCl_2_/mL) rennet by Ets Coquard Co. (Villefranche-sur-Saône, France), were used to denaturate the milk. After processing, all 3 kg cheese pieces were left to drain at room temperature for approximately 5 h. Ripening was carried out in KK350 TOP + POLEKO ripening cabinets ensuring stable conditions of 5 °C, 10 °C, and 15 °C at 90% humidity.

### 3.2. Moisture Content Analysis

We controlled the cheese mass during the ripening process with the WPT2 Radwag laboratory balance. Drying the product in a convection dryer and measuring the dry matter allowed us to determine the moisture content. The moisture ratio MR was calculated according to Equation (1), where M (t) is the actual moisture content, M0 is the initial moisture content, and Me is the equilibrium moisture content. The value of the equilibrium moisture content Me is usually very low, and Equation (1) is often simplified to the form of Equation (2) without significantly changing the value of MR [51,52].
(1)$$\mathrm{MR}\;=\frac{\mathrm{Mt}-\mathrm{Me}}{M0-\mathrm{Me}}$$
(2)$$\mathrm{MR}\;=\frac{\mathrm{Mt}}{M0}$$

### 3.3. GC/MS Analysis

Solid-phase microextraction (SPME) was used to isolate and concentrate the volatile flavor compounds [53]. A representative cheese sample (4.2 g) was transferred to a 20 mL septum-capped vial with 0.123 µg of 2–undecanone as an internal standard. During isolation, samples were placed in an 80 °C water bath with stirring. Each sample was equilibrated and absorbed on the SPME fiber (2 cm–50/30 μm divinylbenzene/carboxen/polydimethylsiloxane; Supelco, Inc., Bellefonte, PA, USA) for 45 min. A Clarus 680 gas chromatograph coupled to a Perkin Elmer mass spectrometry detector (GC/MS) SQ 8S by Perkin Elmer, with a column ELITE-5MS (Crossbond 5% diphenyl and 95% dimethylpolysiloxane) 30 m × 0.25 mm ID × 0.25 μm film was used. Scanning was performed from 50 to 350 m/z in electronic impact (EI) at 70 eV, mode at 5 scan s^−1^. Analyses were performed using helium as the carrier gas at a flow rate of 1.0 mL min^−1^ in a split ratio of 1:10 and the following program: (a) 45 °C for 2 min; (b) rate of 5.0 °C min^−1^ for 45–160 °C; and (c) rate of 15 °C min^−1^ for 160–280 °C. The injector was maintained at 250 °C in each case. Three different analytical methods were employed to perform most of the compound identification: (1) retention indices (KI), (2) GC/MS retention times (authentic chemicals), and (3) mass spectra, with similarity indexes > 90%-authentic chemicals and NIST11 spectral library collection [54]. All aroma standards used for identification and quantification were food grade [55]. Figure 4 shows an example chromatogram of the analysis of aromas of the tested cheeses.

### 3.4. Sensory Analysis

Eight trained testers performed the sensory analysis; each tester was given a whole cheese and a smooth-bladed knife. First, the external appearance features were evaluated. Second, consistency, odor, taste, and salt content were evaluated using a five-point scoring scale (5—liked very much, 1—disliked) [56,57].

### 3.5. Statistical Analysis

Tukey’s test, with the significance level of α = 0,05, was used to measure the ripening effects in the cheeses. In the analysis of volatility, a one-way analysis (ANOVA) was performed to detect significant differences in the concentrations of analytes as a function of the main factors studied (ripening temperature). The data were processed with Statistica (version 10) by StatSoft [58].

## Figures and Tables

**Figure 1 molecules-27-08745-f001:**
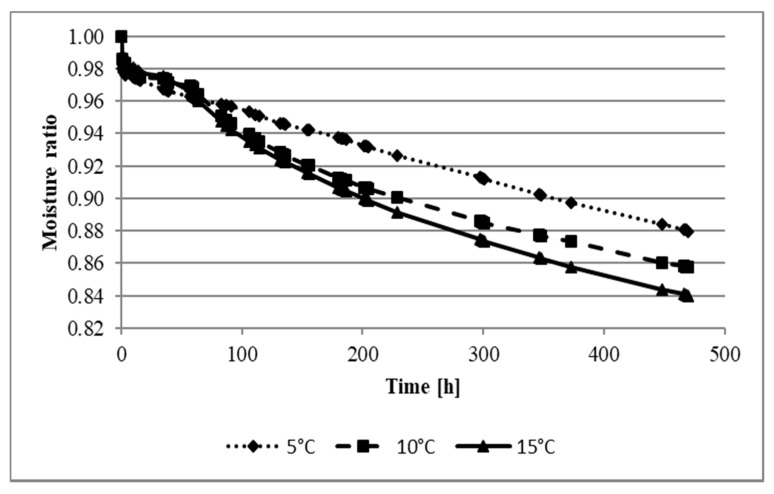
Changes in the moisture ratio of Korycin cheese during ripening at different temperatures.

**Figure 2 molecules-27-08745-f002:**
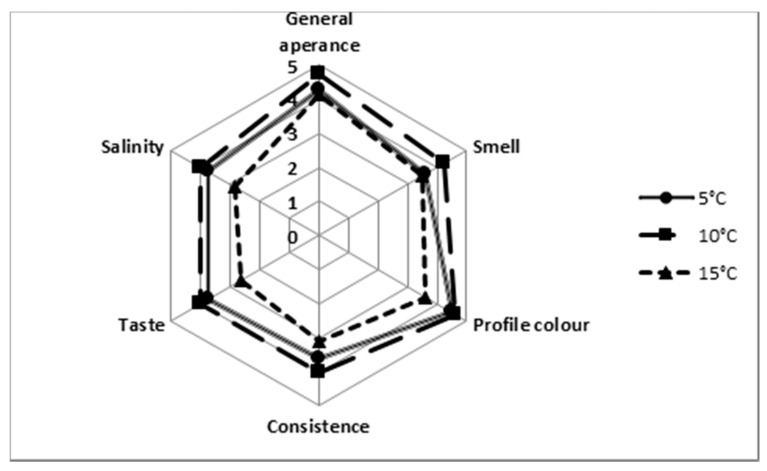
Sensory evaluation scores of Koryciński cheese ripened in various conditions.

**Figure 3 molecules-27-08745-f003:**
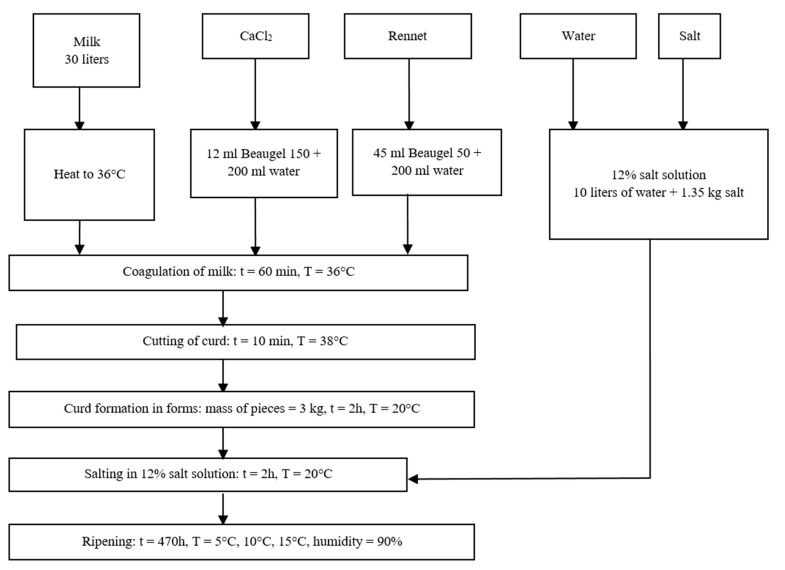
Technological procedures and process conditions in the production of laboratory-scale Koryciński cheese.

**Figure 4 molecules-27-08745-f004:**
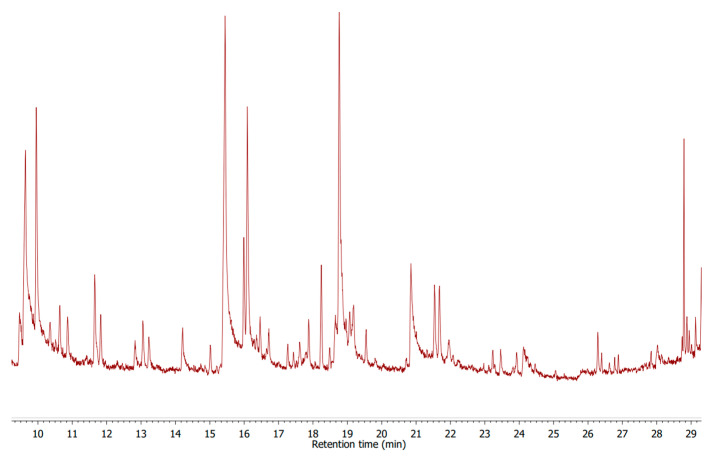
Typical chromatogram of an analysis of the volatile components of cheese by GC/MS.

**Table 1 molecules-27-08745-t001:** Volatile content in analyzed cheese [µg/kg]. The content of chemical compounds in the aroma is marked with the same letters (a, b, c) when they do not differ statistically significantly. Statistically homogeneous groups at α = 0.05 marked with the same letters.

Compound Name	KI Exp	KI Lit	5 °C	10 °C	15 °C
Pentanoic acid	858	858	1.30 ^a^	4.60 ^b^	27.08 ^c^
Heptenal (4Z)	899	901	1.60 ^a^	2.10 ^a^	1.94 ^a^
Hexanoic acid	996	996	26.23 ^a^	69.70 ^b^	108.94 ^c^
Hexanoic acid, ethyl ester	1000	1000	17.28 ^a^	5.40 ^b^	10.79 ^a^
Hexyl acetate	1015	1007	1.35 ^b^	1.54 ^b^	0.87 ^a^
Nonane, 2,6–dimethyl-	1025	1025	2.48 ^c^	1.18 ^b^	0.61 ^a^
Limonene	1032	1033	2.45 ^c^	1.09 ^b^	0.42 ^a^
4–Hexanolide	1059	1057	6.68 ^c^	3.57 ^b^	2.28 ^a^
2–Nonanone	1092	1092	3.48 ^b^	1.33 ^a^	6.15 ^c^
n–Undecane	1100	1100	2.87 ^b^	3.34 ^c^	0.53 ^a^
2–Nonen–1– al	1140	1144	2.94 ^b^	3.57 ^b^	1.60 ^a^
Octanoic acid	1183	1180	43.91 ^a^	84.90 ^b^	103.54 ^c^
Octanoic acid, ethyl ester	1197	1198	6.28 ^c^	2.89 ^a^	5.47 ^b^
n–Dodecane	1200	1200	16.35 ^c^	8.33 ^b^	4.37 ^a^
Undecane, 2,6–dimethyl–	1214	1210	1.96 ^b^	1.45 ^b^	0.04 ^a^
Methyl nonanoate	1223	1224	1.14 ^c^	0.65 ^b^	0.04 ^a^
Hexadienol butanoate <(2E,4E)–>	1243	1248	1.42 ^c^	0.74 ^b^	0.04 ^a^
Unknown	1264		2.43 ^c^	1.27 ^b^	0.04 ^a^
Dodecane, 2,6,11–trimethyl–	1276	1275	6.66 ^c^	3.42 ^b^	1.86 ^a^
2–Undecanone	1293	1293	29.29 ^a^	29.29 ^a^	29.29 ^a^
Nonadienol acetate <(2E,6Z)–>	1304	1304	1.54 ^b^	1.06 ^b^	0.04 ^a^
methyl Decanoate	1322	1322	1.68 ^b^	0.00 ^a^	0.04 ^a^
n–Decanoic acid	1372	1363	10.53 ^a^	28.13 ^b^	40.00 ^c^
n–Decanoic acid, ethyl ester	1395	1410	4.88 ^b^	2.57 ^a^	2.96 ^a^
Tetradecane	1400	1400	4.62 ^b^	4.61 ^b^	1.82 ^a^
Dodecanal	1411	1408	0.02 ^a^	1.86 ^b^	3.57 ^c^
Benzyl acetalacetate	1488	1486	1.03 ^b^	1.48 ^b^	0.04 ^a^
Pentadecane	1494	1500	0.02 ^a^	6.17 ^b^	7.10 ^b^
Dodecanoic acid, ethyl ester	1595	1594	2.20 ^b^	0.89 ^a^	0.99 ^a^
Hexadecane	1599	1600	0.61 ^a^	0.80 ^a^	0.49 ^a^
Camphoric acid	1634	1634	0.02 ^a^	0.38 ^b^	0.53 ^b^
Heptadecane	1699	1700	0.02 ^a^	0.12 ^a^	0.30 ^a^
δ–Dodecalactone	1712	1719	0.02 ^a^	0.53 ^b^	2.28 ^c^
n–Pentadecanol	1786	1773	7.43 ^b^	3.25 ^a^	4.44 ^a^
1–Decanol, 2–hexyl–	1793	1795	1.40 ^a^	7.76 ^b^	0.46 ^a^
n–Octadecane	1799	1800	0.77 ^a^	0.77 ^a^	0.46 ^a^
Hexadecanal	1807	1815	0.02 ^a^	0.27 ^b^	0.27 ^b^
Tetradecanal	1819	1819	1.17 ^a^	1.09 ^a^	0.87 ^a^
2–Hexadecene, 2,6,10,14–tetramethyl–	1844	1849	2.73 ^b^	2.78 ^b^	1.63 ^a^

## Data Availability

The data are available at the research unit of the Institute of Dietetics and Dietotherapy, Faculty of Biological Sciences-University of Zielona Góra, Kalsk 67, 66–100 Kalsk. Or made available upon direct request: j.kliks@wzs.uz.zgora.pl.

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
