# Peer review of "The Aroma Composition of Koryciński Cheese Ripened in Different Temperatures"

_molecules, 2022, doi:10.3390/molecules27248745_

Round 1

Reviewer 1 Report

Review:

Shaping the aroma composition of Koryciński cheese by ripening in controlled conditions by Kliks et al.

The authors describe the manufacture, ripening and analysis of volatile compounds in Korycin cheese. The effect of three experimental ripening temperatures on the development of volatile components were compared. The development of some components of the major chemical groups: esters, aldehydes, organic acids and lactones were shown to develop differently.

The subject is a village cheese, which must be a derivative of some larger group of cheeses. During the 1990’s researchers described the chemical changes during ripening of the major cheese groups such as cheddar (salted hard), gouda (hard), brie (mold ripened) etc. This was followed by similar work on village cheeses. In the results and discussion section the authors briefly mention that: “certain changes were also noted for other cheeses [reference]”. If the authors consider their work of useful value, it would be important that the larger category in which the Korycin cheese falls is taken into account and that the comparisons they refer to are made with specific cheeses whether the latter fall in the same category as the Korycin cheese.

The document needs technical editing and proper English editing. Below are a few examples from the many errors detected in almost every line on the first two pages.

Title: what is meant by “shaping” of the aroma composition. This word could be omitted from the title without changing the meaning.

Line 20: Fromm 2005 its…

21: …in 2012 it…

23: …municipalities of the Podlaise province located…

24: The community of…

25: …in 2011 it united over…

26: …with the use of rennet and temperature of denaturation of 33-36…

27: …dependent on…

28: …weeks at room temperature, which…

29: The popularity of the Korycin … owed to those…

31: Many biological and chemical processes are occurring during…

33: …and proteolysis generally have autocatalytic characteristics as opposed to the…connected with microbial…

35: In general the raw milk microbial composition consists of:

36-40: Latin names should be in Italic font.

48: Figure 3 shows…

60: Figure 4 shows…

The figures should be in chronological order.

Change titles of Figures. Suggested:

Figure 3. Changes in moisture ratio of Korycin cheese during ripening at different temperatures.

Figure 4. Sensory scores of Korycin cheese ripened at different temperatures.

82, 87, 89: When directly referring to other researchers write as follows: …shown by Atik et al. [4] and Bertuzzi et al. [5].

Table 1. Content of volatile components of Korycin cheese [ug/kg] ripened at different temperatures.

In the table, group the components in the major groups that were mentioned in results and discussion under sub-headings in the column at the left: esters, aldehydes, organic acids, lactones.

Regarding the statistical reporting in Table 1: exchange commas with full-stops – 1,30 change to 1.30.

The letters indicating statistical significance should be super scripts.

Author Response

Dear Reviewer,

Please, find enclosed our manuscript entitled:

“The aroma composition of KoryciÅ„ski cheese ripened in diferent temperatures"

We would like to thank the reviewers for their valuable comments and contributions. Your advice allowed us to improve the substantive value of our article. We hope that we have managed to address all the comments you have given us. Below we will allow ourselves to refer to individual comments and indicate the corrections we have made.

Response to Reviewer 1:

Point 1:

„In the results and discussion section the authors briefly mention that: “certain changes were also noted for other cheeses [reference]”. If the authors consider their work of useful value, it would be important that the larger category in which the Korycin cheese falls is taken into account and that the comparisons they refer to are made with specific cheeses whether the latter fall in the same category as the Korycin cheese.”

Response 1:

Koryciński cheese is difficult to unambiguously classify due to the regional nature of production, but it should be assigned to the semi-hard cheese group. Researchers in the results and discussion section pointed out similarities in the ripening of similar cheeses.

Point 2:

“The document needs technical editing and proper English editing. Below are a few examples from the many errors detected in almost every line on the first two pages.”

Response 2:

The document has been professionally edited by a native speaker from University od Zielona Gora.

Point 3:

“Title: what is meant by “shaping” of the aroma composition. This word could be omitted from the title without changing the meaning.”

Response 3:

The title has been corrected.

Point 4:

Line 20: Fromm 2005 its…

21: …in 2012 it…

23: …municipalities of the Podlaise province located…

24: The community of…

25: …in 2011 it united over…

26: …with the use of rennet and temperature of denaturation of 33-36…

27: …dependent on…

28: …weeks at room temperature, which…

29: The popularity of the Korycin … owed to those…

31: Many biological and chemical processes are occurring during…

33: …and proteolysis generally have autocatalytic characteristics as opposed to the…connected with microbial…

35: In general the raw milk microbial composition consists of:

36-40: Latin names should be in Italic font.

48: Figure 3 shows…

60: Figure 4 shows…

The figures should be in chronological order.

Change titles of Figures. Suggested:

Figure 3. Changes in moisture ratio of Korycin cheese during ripening at different temperatures.

Figure 4. Sensory scores of Korycin cheese ripened at different temperatures.

82, 87, 89: When directly referring to other researchers write as follows: …shown by Atik et al. [4] and Bertuzzi et al. [5].

Table 1. Content of volatile components of Korycin cheese [ug/kg] ripened at different temperatures.

In the table, group the components in the major groups that were mentioned in results and discussion under sub-headings in the column at the left: esters, aldehydes, organic acids, lactones.

Regarding the statistical reporting in Table 1: exchange commas with full-stops – 1,30 change to 1.30.

The letters indicating statistical significance should be super scripts.

Response 4:

Thank you for pointing out the remaining errors in detail and for very specific suggestions for their improvement. We hope that we have managed to include all of them in the text.

Reviewer 2 Report

This manuscript mainly focused on the aroma composition of Koryciński cheese during ripening process, and tired to elaborate the relationship between organoleptic properties and ripening condition. This could be an interesting study, however, there are several defects existed in the present format. The specific points were as follow:

1. It is difficult to see the novelty of the research from the current manuscript format.

2. It is sure that the aging temperature had a key influence on the cheese formation. However, the factors as temperature in other procedures, aging time, or even additive type and dosage etc. also had effect on the cheese characteristics. Thus, the experiments designed in this study could not support the title “Shaping the aroma composition of KoryciÅ„ski cheese by ripening in controlled conditions”.

3. The writing of the manuscript lacks of academic rigor. For instance, the introduction of materials as milk and other reagents lacks the detailed product information.

4. The introduction lacks the the latest research progress on the influence of cheese flavor substances on its sensory properties.

Author Response

Dear Reviewer,

Please, find enclosed our manuscript entitled:

“The aroma composition of KoryciÅ„ski cheese ripened in diferent temperatures"

We would like to thank the reviewers for their valuable comments and contributions. Your advice allowed us to improve the substantive value of our article. We hope that we have managed to address all the comments you have given us. Below we will allow ourselves to refer to individual comments and indicate the corrections we have made.

Response to Reviewer 2:

Point 1:

„It is difficult to see the novelty of the research from the current manuscript format.”

Response 1:

The key aspect indicating the attractiveness of these studies is the endemic nature of the product itself, which occurs only locally in Poland in the Podlasie region. So far, no studies have been published that would allow for a detailed study of the volatile profile of these unique cheeses. In their work, the researchers undertook to participate in the production process and transferred some of the material to controlled laboratory conditions using ripening chambers. In addition, the researchers were the first to organize panel studies aimed at determining the organoleptic parameters of Koryciński cheeses. The correlation of the results of organoleptic tests with the aromatic profile makes it possible to indicate compounds that are of key importance for the quality of these cheeses.

Point 2:

„ It is sure that the aging temperature had a key influence on the cheese formation. However, the factors as temperature in other procedures, aging time, or even additive type and dosage etc. also had effect on the cheese characteristics. Thus, the experiments designed in this study could not support the title “Shaping the aroma composition of KoryciÅ„ski cheese by ripening in controlled conditions”.”

Response 2:

Thank you for your attention, the title has been changed to reflect the profile of the experiment.

Point 3:

„The writing of the manuscript lacks of academic rigor. For instance, the introduction of materials as milk and other reagents lacks the detailed product information.”

Response 3:

Information on the parameters of the milk used in the study was corrected and the producers and origin of technological additives were supplemented.

Point 4:

”The introduction lacks the the latest research progress on the influence of cheese flavor substances on its sensory properties.”

Response 4:

The introduction has been supplemented and the current review of scientific achievements in the field of shaping the aroma profile of cheeses and research techniques.

Round 2

Reviewer 1 Report

The requested changes were made to the document.

An extensive english editing was carried out.

The document may be published.

Author Response

Dear Reviewer,

Please, find enclosed our manuscript entitled:

“The aroma composition of KoryciÅ„ski cheese ripened in diferent temperatures"

Response to Reviewer 1:

Point 1:

The requested changes were made to the document.

An extensive english editing was carried out.

The document may be published.

Response 1:

We would like to thank the reviewer for his comments and final assessment of our work.

Reviewer 2 Report

I'm glad to see that the author has revised the manuscript, and this format has been improved a lot. It is recommended that the author further check the article, improve the writing, and try to quote the latest references.

Author Response

Dear Reviewer,

Please, find enclosed our manuscript entitled:

“The aroma composition of KoryciÅ„ski cheese ripened in diferent temperatures"

We would like to thank the reviewers for their valuable comments and contributions. Your advice allowed us to update the state of the art in our article, which should make it more scientifically valuable. We hope that the article will meet the publication requirements.

Response to Reviewer 2:

Point 1:

„I'm glad to see that the author has revised the manuscript, and this format has been improved a lot. It is recommended that the author further check the article, improve the writing, and try to quote the latest references.”

Response 1:

Thank you for your comments, which had a significant impact on improving the quality and relevance of our article. We removed old articles from the text, updated 11 source articles, finding more up-to-date data to which we referred. The article has been re-edited to improve the style.
